# The Edible Gray Oyster Fungi *Pleurotus* *ostreatus* (Jacq. ex Fr.) P. Kumm a Potent Waste Consumer, a Biofriendly Species with Antioxidant Activity Depending on the Growth Substrate

**DOI:** 10.3390/jof8030274

**Published:** 2022-03-09

**Authors:** Raluca A. Mihai, Erly J. Melo Heras, Larisa I. Florescu, Rodica D. Catana

**Affiliations:** 1CICTE, Department of Life Science and Agriculture, Universidad de Las Fuerzas Armadas—ESPE, Av. General Rumiñahui s/n y, Sangolqui 171103, Ecuador; ejmelo@espe.edu.ec; 2Taxonomy, Ecology and Nature Conservation Department, Institute of Biology Bucharest of Romanian Academy, 296 Splaiul Independentei, 060031 Bucharest, Romania; larisa.florescu@ibiol.ro; 3Developmental Biology Department, Institute of Biology Bucharest of Romanian Academy, 296 Splaiul Independentei, 060031 Bucharest, Romania; rodica.catana@ibiol.ro

**Keywords:** waste, recyclable substrates, oyster fungi, antioxidants, phenolic compounds

## Abstract

Nowadays, climate change is not the only threat facing our planet. There are also other types of pollution such as waste that poisons soils and water and kills plants, harming humans and animals. Sustainability represents a key issue for the actual Global Citizen. For this reason, our article is dedicated to offering biofriendly solutions to decrease wastes, give them a positive meaning, such as a substrate for an edible oyster fungus with nutritive and biological properties usefully for humans. Three types of wastes such as coconut coir, pine sawdust, and paper waste—representative symbols of pollution in Ecuador—have been tested as suitable growing substrate for the edible fungi *Pleurotus ostreatus* (Jacq. ex Fr.) P. Kumm by analyzing parameters such as *Biological Efficiency*, *Mushroom Yield*, *and Productive Rate*. The influence of these “waste” substrates on the nutritive (protein content), biological characteristic (antioxidant activity), and the content of human-health-sustaining compounds (phenols, flavonoids) were also evaluated using the Kjeldahal, DPPH, ABTS, FRAP, and Folin–Ciocalteu methods. The results indicate that all the waste products represent desirable substrates for growing the edible fungi, with more focus on coconut coir waste (one of the principal pollution problems in Ecuador), but that also achieved the increase in the fungi’s desirable characteristics. Coconut coir waste could be an environmentally friendly solution that also offers for humans additional nutritive and healthy benefits.

## 1. Introduction

All over the world, different types of wastes are generated from different industries and household waste, etc. Since 2014, the waste issue has been recognized as a global environmental problem [1]. An inefficient waste management contributes to air pollution, affecting the health and well-being of ecosystems, species, and humans. Recycling plays important role in finding a sustainable solution, with low impact on the environment, which improves the circular economy and protects natural resources.

An immense amount of waste was produced in the last years due to the intensification of agricultural and industrial activities [2], being estimated at ~998 million tons/year (only the agricultural waste) [3]. Increasing and improper disposal of agro-industrial waste becomes a major source of pollution, affecting the population and environment health and amplifying the global emissions of greenhouse gasses [4]. The use of agro-industrial waste as raw materials for various products (antioxidants through solid-state fermentation) may help to reduce the cost of production, reducing the environmental pollution [5]. Reduction in greenhouse gas emissions, bio-energy, bio-conversion, new jobs, and new green markets are some benefits of recycling agricultural solid wastes, which encourages future studies [6].

In tropical areas, a large amount of unused lignocellulosic by-products is available, being left to rot in the field or burned [7]. Ecuador is one of those areas which has an ideal climate for coconut cultivation, Esmeraldas being the province with the highest coconut production (77.26%), followed by Manabí (18.72%). In Esmeraldas, the cultivation of coconut is concentrated in the border cantons: Eloy Alfaro and San Lorenzo del Pailón [8]. Coconut residue is a by-product waste rich in beneficial nutrients and available in high quantities in Ecuador, without cost [9]. Due to its alimental characteristics, the fiber and shell of the coconut represent wastes with great potential as a component of growing substrates for edible mushrooms [10]. Despite its nutritional benefits (0.52% protein, 0.29% fat, 35.44% fiber, 3.94% ash, and 20.53% humidity, low conductivity, resistance to impact and bacteria [11]), the use of coconut waste in the country is almost null, being considered an environmental problem [12].

Pine wood agroindustry or by-products wastes are suitable for bio-energy production but are difficult to use due to their chemical composition [13], so an alternative for these wastes can be as a substrate for the culture of *Pleurotus* [14], scarcely applied at the moment. At the same time, the pulp and paper industry occupies the place of the third largest industrial polluter of the environment (air, water, and soil). Chlorine-based bleaches used during paper production result in toxic materials being released into the environment. In addition, methane is formed during the rotting of paper waste, and is 25 times more toxic than CO_2_. At the moment, the volume of global waste paper has reached 1010 million tons in the last 25 years [15].

In the context of the high amount of different waste sources (coconut, wood, paper), the cultivation of *P. ostreatus* on these waste by-products may be one of the solutions to transform the inedible waste into an edible rich in proteins and with high antioxidant character biomass of high market value [7].

Fungi are of interest in various bioremediation processes due to their ability to grow and develop in different substrates (natural and synthetic solid materials) [16]. Due to their ligninocellulite enzymes [17], fungi are able to transform the lignocellulosic biomass represented by agricultural and forestry wastes into food [18]. This capacity has a high potential application in most industries (paper, chemical, textile, food), agricultural processes, forage production [19], and bioenergy production from agro-industrial waste [20].

Mushroom cultivation represents a useful method to manage environmental pollution. The oyster fungus grows on different materials as by-products or waste from agricultural and agro-industrial activities, which brings an environmental benefit due to the importance of waste management [18]. The genus *Pleurotus* is recognized as species with high nutritional value, having various biotechnological and environmental applications [16]. *Pleurotus* species represent not only a commercially important edible mushroom [21] but also an important component in recycling agricultural wastes turning into protein-rich food. Due to their attractive taste and aroma, nutritional and medicinal value, these species are cultivated globally using numerous agricultural by-products (banana, corn, sugarcane leaves, peanut hull, wheat, rice straw, mango fruits and seeds, leaves) with low-cost production techniques [3]. *Pleurotus* species are considered an important source of dietary fiber, contain numerous important nutrients and polyphenols, which assure their antioxidant character and ability to inhibit free radicals [22]. Although the human body is a balance between free radicals and antioxidants, it is still necessary as a dietary antioxidant to reduce the oxidative stress from the environment [23]. These mushrooms are considered a cheap source of protein since they convert agricultural waste into food [24].

*Pleurotus ostreatus* is member of the *Pleurotus* genus and represents the second most cultivated edible mushroom after *Agaricus bisporus* due to its economic (edible), ecological (bioremediation agents), and medicinal value (antioxidant activity and biocompounds source) [25]. This fungus is able to colonize a large variety of lignocellulosic substrates and other agricultural, forest, and food-processing wastes [26], and its cultivation is an effective alternative to produce valuable food and nutraceuticals. As a member of basidiomycetes group, it brings to light the presence of ribotoxin—like protein (Ostreatin), a novel specific ribonucleases family. Ostrein could contribute to assigning potential biotechnological applications in agriculture (crops protection towards pathogens or pests) and in medicine (cytotoxic effects) [27].

The advantage of *P. ostreatus* is that it has a shorter growth time (comparing with other edible mushrooms), the substrate used requires pasteurization, it has high profitability (converting a high percentage of the substrate to fruiting bodies), and it is less attacked by diseases and pests [26].

Although there are some studies concerning the cultivation of *P. ostreatus* on different substrates (chestnut, corncobs, beech, oak, linden, potatoes farm wastes, walnut, poplar, peanut wastes, walnut and orange tree sawdust, etc.), to our knowledge, there are still limited studies about the antioxidant character of *P. ostreatus* cultivated on recyclable substrates such as pine wood, coconut coir (fiber, shells), and waste paper.

The study was conducted to evaluate different agricultural wastes (pine sawdust, coconut coir, and paper waste), representative of contamination for Ecuador, as suitable opportunity to diminish the pollution by their recycling into growing substrate for a bio- friendly fungi with desirable organoleptic, phytochemical properties, and antioxidant activity, with final human health and protective planet benefits.

## 2. Materials and Methods

### 2.1. Study Area, Experimental Substrates, and Spawn Preparation

Three different contaminant wastes (pine sawdust—T1; coconut coir—T2; and waste paper—T3) were evaluated for biological efficiency, yield, production rate, and antioxidant activity of oyster mushroom. In our study, coconut coir was represented by the mixture of fiber and sawdust from dried coconut shells, collected from local coconut plantations in Esmeraldas, localities Borbón, Limones. Paper waste was collected from the administrative offices of the University, and pine sawdust was purchased from wood factories. It was taken into account that each substrate must have a humidity lower than 12% for its use, securing a predominantly dry and mold-free material [28]. The substrates were ground into 2–5 cm length pellets and soaked in water at 60 °C for 24 h. Excess water was drained off from the materials, mixed with 10% wheat bran, and 2% gypsum (calcium sulfate) as nutrient supplement, at pH 9 [29]. Water content was attained until optimum moisture (75%). The mixture was then filled into 9′ × 14′ polystyrene bags that were pasteurized for 6 h to eliminate bacteria or other contaminant fungi that would compete with the oyster mushroom for the substrate.

Pure culture of oyster mushroom (*P. ostreatus*) was obtained from the mushroom farm “Fungi Andino” located in the neighborhood La Morita, Tumbaco Valley in Quito, Ecuador. Spawns were prepared in 1 kg polystyrene plastic bags filled with 600 g of oatmeal kernel, which was hydrated with 75% of water. Each bag was supplemented with 2% gypsum, in terms of dry weight basis and pasteurized for 6 h. After cooling at room temperature, each sterilized bag was inoculated with 250 g of primary mycelium. The spawn was incubated for 28 days at 25 °C.

#### 2.1.1. Inoculation, Incubation, and Harvest

The sterilized bags were then allowed to cool at room temperature. The inoculation took place in a tissue culture hood to avoid contamination, so the bags were aseptically inoculated with a piece, 10% in weight, of mycelium culture of 14 days old. The bags were subsequently incubated at 21 ± 3 °C for 28 days until the mycelium fully invaded the substrate and then relocated to a fruiting room with light entry with a mean of 12 h of light per day. After mycelium growth in the bags became abundant, to facilitate the development of fruiting bodies, perforations were created in the bags. Then, the fully colonized substrates were transferred to the growth room and placed on racks at a spacing of 15–20 cm. Proper ventilation of the growth room was assured by opening the door every 2–3 days. To keep the mycelium moist, the inoculated bags were watered 2–3 times a day. Relative humidity (RH) and room temperature were monitored and maintained with thermo-hygrometer hydrometer, and RH was maintained between 80 and 85% by spraying fine mist of water occasionally [30].

#### 2.1.2. Biological Efficiency, Mushroom Yield, and Production Rate

The growth and development of mushrooms were monitored daily. The time (number of days) required from inoculation to completion of mycelium running, time elapsed between opening the plastic bags to pinhead formation, and time required from opening the plastic bags to first round harvesting were recorded. One round of mushroom harvest was made across all substrate types in the course of the experiment.

To evaluate the growth performance of mushrooms on different substrates, yield, biological efficiency, and production rate were calculated for each experimental treatment. The mushroom yield (%) was determined by weighing the total amount of harvested fruiting bodies without taking the base out, which was then divided by the total weight of the dry substrate used. Biological yield (g) was determined by weighing the whole cluster of fruiting bodies without removing the base of stalks, and economic yield (g) was determined by weighing all the fruiting bodies on a substrate after removing the base of stalks. Finally, biological efficiency (%) was calculated by using the following equation:(1)$$\%\;BE=\frac{FW_{m}}{DW_{s}}*100\%$$
where *BE*—the Biological Efficiency (%); *FW_m_*—the fresh weight (g) of the harvested mushrooms; and *DW_s_*—the dry weight of the substrate (g) [31].

The production rate (%) was determined by dividing the biological efficiency (%) between the total number of days of the process [TP = EB (%)/number of days of the process] [32].

#### 2.1.3. Preparation of the Fungi Solution

The fresh oyster *Pleurotus* were cleaned with distilled water before oven drying at 40 °C. When heating to constant weight was achieved, the dried material was grinded in a laboratory mill, and 1.0 g was critically weighed and extracted in an ultrasonic cleaner at 50 °C with 40 times of 80% methanol. The solution was then filtered through a Whatmann filter, and the filter extract was concentrated into a dry powder by the rotary evaporator at 50 °C, dissolved in 70% ethanol, and placed in 25 mL volumetric flasks for further antioxidant and metabolic content analysis.

#### 2.1.4. Determination of the Antioxidant Aptness

The DPPH free radical scavenging activity of fungi samples were determined according to the method described by [33]. The DPPH solution was prepared as reported by [34]. After weighting 7.89 mg DPPH on a chemical balance, it was dissolved in 99.5% ethanol to obtain a constant volume by filling 100 mL of a measuring flask (0.2 mM DPPH). The formed solution was kept in the dark for 2 h until the absorbance was stabilized. After this, 1 mL of DPPH solution was added into a test tube, followed by 200 µL of ethanol and 800 µL of 0.1 M Tris·HCl buffer (pH 7.4). After mixing, the absorbance at 517 nm was measured. A mixed solution containing 1.2 mg of ethanol and 800 µL of Tis·HCl buffer was used as a blank. When the absorbance was in a range of 1.00, the prepared solution was used directly for the measurements. If the absorbance exceeded 1.05, ethanol was added to dilute the solution until the absorbance was in the range of 1.00. The solution used for measurements was stored in the dark during the assay. Briefly, the DPPH Radical Scavenging Assay consists of adding an aliquot (40 μL) of fungi extract to 3 mL of methanolic DPPH solution. The change in absorbance at 515 nm was measured after 30 min, and the antiradical activity (AA) was determined using the following formula:AA% = 100 − [(Abs: sample − Abs: empty sample)] × 100)/Abs: control(2)

The optic density of the samples, the control and the empty samples, were measured in comparison with methanol. One synthetic antioxidant represented by Trolox was used as positive control. The antioxidant capacity based on the DPPH free radical scavenging ability of the extract was expressed as μmol Trolox equivalent per gram of dry weight of fungi material.

#### 2.1.5. ABTS Free Radical Scavenging Assay

The ABTS radical cation scavenging activity was performed according to [35], with slight modifications. The ABTS solution (7 mM) was reacted with potassium persulfate (2.45 mM) solution and kept overnight in the dark to yield a dark-green-color solution containing ABTS radical cation. Prior to use in the assay, the ABTS radical cation was diluted with 50% methanol for an initial absorbance of about 0.700 ± 0.02 at 734 nm using UV-VIS spectrophotometer. Free radical scavenging activity was assayed by mixing 100 µL of test sample with 2.9 mL of an ABTS working standard in a microcuvette. The decrease in absorbance was measured at exactly 1 min after mixing the solution and then at 1-min intervals up to 6 min, when final absorbance was recorded. The inhibition % was calculated using the formula: Inhibition% = A (control) − A (test sample)/A (control) × 100.

The antioxidant radical scavenging activity in the mushroom extract is evaluated using DPPH and ABTS radicals. The absorbance was measured against the reagent blank at 515 nm for DPPH and 435 nm for ABTS. Three repetitions were performed. The radical scavenging activity is calculated according to the regression equations for DPPH (y = −0.8033x + 0.8131, r^2^ = 0.9542) and ABTS (y = −0.4202x + 0.7797, r^2^ = 0.9903) obtained from the TROLOX calibration curves.

#### 2.1.6. Reducing Ability (FRAP Assay)

The ability to reduce ferric ions was measured using a modified method of [36]. An aliquot (200 μL) of the fungi extract with appropriate dilution was added to 3 mL of FRAP reagent (10 parts of 300 mM sodium acetate buffer at 3.6 pH, 1 part of 10 mM TPTZ solution, and 1 part of 20 mM FeCl_3_ 6H_2_O solution), and the reaction mixture was incubated in a water bath at 37 °C. The increase in absorbance at 593 nm was measured after 30 min. The antioxidant capacity based on the ability to reduce ferric ions of the extract was expressed in μM Fe (II)/g dry mass and compared with ascorbic acid as standard.

The Fe^2+^ calibration curve for reducing activity was used to calculate the Fe^2+^-TPTZ concentration, which reveals the existence of metabolites with antioxidant capacity in the mushroom extract. The calibration curve was calculated, obtaining the following equation: y = 1.5431x + 0.0004 (*n* = 3, r^2^ = 0.9961).

#### 2.1.7. Determination of Total Phenolic Content (TPC)

Obtaining the TPC content of our fungi extracts was achieved using the methodology described by [37], based on the Folin–Ciocalteu method. Briefly, 10 mg of gallic acid was dissolved in 100 mL of 50% methanol (100 μg/mL) and then further diluted to 6.25, 12.5, 25, or 50 μg/mL [38]. Next, 1 mL aliquots of each dilution were taken in a test tube and diluted with 10 mL of distilled water. Then, 1.5 mL Folin–Ciocalteu’s reagent was added and incubated at room temperature for 5 min; 4 mL of 20% (*w*/*w*) Na_2_CO_3_ was added to each test tube, adjusted with distilled water up to the mark of 25 mL, agitated, and left to stand for 30 min at room temperature. Absorbance of the standard was measured at 765 nm using UV/VIS spectrophotometer, with distilled water as blank. Total phenolic content was expressed as gallic acid equivalent (GAE) in the dry sample. Results were expressed percentage *w*/*w* and calculated using the following formula: Total phenolic content (% *w*/*w*) = GAE × V × D × 10^−6^ × 100/W, where GAE—Gallic acid equivalent (μg/mL); V—Total volume of sample (mL); D—Dilution factor; and W—Sample weight (g).

For the calculation of the TPC values, an experimental calibration curve was taken with the equation y = 0.0157x + 0.0357 (*n* = 3, r^2^ = 0.9968), which was obtained for gallic acid, where y represents the known pyrogallol concentration and x is the registered absorbance.

#### 2.1.8. Total Flavonoid Content Determination

The quantification method for total flavonoids contents followed the method in [36]. The sample (1 mL) consisting of the fungi extract was mixed with NaNO_3_ (0.3 mL) in a test tube covered with aluminum foil and left for 5 min. Then, 10% AlCl_3_ (0.3 mL) was added, followed by 1 M NaOH (2 mL). Later, the absorbance was measured at 510 nm using a spectrophotometer with quercetin as a standard (results expressed as mg/g^−1^ quercetin dry sample).

#### 2.1.9. Protein Content Determination

Fresh oyster mushroom samples were cleaned of substrate residues, then dried in a food dehydrator at a low temperature of 40 °C and ground to a fine powder before sifting to remove lumps. This procedure is necessary to prevent lignin artifacts in the powder [39]. The crude protein content in dried mushrooms was determined by the macro-Kjeldahl method using the conversion factor of N × 4.38 [40] and using 3 repetitions for each treatment.

#### 2.1.10. Sensory Evaluation and Organoleptic Properties

The organoleptic properties and sensory evaluation of the oyster mushroom fruiting bodies were conducted through a preference test. The three treatments were assessed by means of 7 sensory attributes and 5 acceptability parameters using the 15 cm labelled magnitude scales (LMS), described in the Table 1. Thirty untrained participants tasted the fruiting bodies following a randomized order and were asked to assess the intensity of each attribute. Acceptability parameters were also evaluated by the same LMS preference test described in Table 1. The evaluated sensory attributes and consumer test were selected by the studies of [41,42], respectively, while acceptability parameters were selected by following the preference test conducted by [43].

### 2.2. Statistical Analysis

The experiment was conducted on a completely randomized experimental design of one factor with 3 treatments and 20 repetitions each. The statistical hypotheses were evaluated according to the proposed experimental design, where the means of the treatments are compared in a one-way analysis of variances (ANOVA). The statistics analysis was determined using the statistics software [44].

The one-way analysis of variance (ANOVA) and Tukey post hoc test were performed for the comparison of treatments. The principle of the method allows for a comparison between the averages of two or more data sets, based on the same principles as the Student’s *t* test. Tukey post hoc test provides additional information to ANOVA analysis, highlighting the significance of differences between pair groups.

Pearson’s correlation matrix was applied to establish the relationships between the variables. Our results contain Pearson’s correlation coefficient r and *p*-values, presented in the same table. It is a fast method to highlight simultaneously, through a matrix, the interdependence relations between N variables.

The statistical significances of *p*-value for both the ANOVA and Pearson correlation matrix are classified as follows: *p* > 0.05 is not statistically significant and statistically significant for *p* ≤ 0.05 (*), *p* ≤ 0.01 (**); *p* ≤ 0.001 (***) and *p* < 0.0001 (****).

An easy visualization of the correlations between the variables was obtained by principal component analysis (PCA). The PCA can establish the relationships of a large number of variables depending on the principal components (axes or factors) resulting in the analysis. The first identified component (F1) is assigned to the largest of the data variants. The second component corresponds to the second variant. The analysis was performed with [45].

## 3. Results

### 3.1. Biological Efficiency, Mushroom Yield, and Production Rate

The biological efficiency, mushroom yield, and production rate were measured for the first fruiting. An analysis of variance was performed to determine the differences between treatments. The statistical analysis revealed that there were significant differences (*p* ≤ 0.05) in the biological efficiency and yield of *P. ostreatus*. The best values for the biological efficiency and yield of *P. ostreatus* were obtained on the waste paper substrate variant, almost double than the value obtained in the pine sawdust (Figure 1).

### 3.2. Antioxidant Aptness

#### Antioxidant Capacity through DPPH and ABTS Tests

The statistical analysis revealed significant differences between means (*p* < 0.05) (Figure 2). Similar results are observed between treatments in respect of the other tests. The coconut coir substrate (T2) presented the highest antioxidant activity, followed by the pine sawdust (T1) and, finally, the waste paper (T3), with a value similar to that of pine sawdust.

### 3.3. Reducing Activity

An ANOVA test was conducted to compare the treatments’ means, obtaining significant differences between the three treatments (*p* < 0.05). In Figure 3, the results of the test are observed. A higher value for treatment 2, represented by the coconut coir substrate, was obtained in comparison with T1 and T3 that showed almost similar results.

### 3.4. Determination of the Total Phenolic Content (TPC)

Comparing the means reveals a significant difference in the values between the different treatments (*p* < 0.05). The samples’ growth on the coconut fiber (T2 substrate) showed a high total polyphenol content (586.60 ± 31.97 mgGAE/g dw), followed by T1 and T3 with close results (397.22 ± 14.87 mgGAE/g dw and 302.95 ± 19.40 mgGAE/g dw) (Figure 4). A One-way ANOVA (F = 38.64, *p* = 0.000) showed significant differences between treatments. The Tukey post hoc test established that the test response was mainly determined by the differences between T1 vs. T2 (*p* = 0.003) and T2 vs. T3 (*p* = 0.0005). The post hoc test confirmed that between T2 and T3 there was no significant differences.

### 3.5. Determination of Total Flavonoid Content

The highest content of total flavonoids in our samples was found in samples cultivated on coconut fiber (T2 variant) (77.54 ± 16.22 mg QE/g dw) being almost double than the total flavonoids content of the samples from recycling paper (T3 variant) (50.79 ± 7.30 mg QE/g dw) (Figure 5). Although a higher value was reached in the T2 treatment, this did not significantly influence the experimental results. Using One-way Anova (F = 0.864, *p* = 0.467), no significant differences were identified between flavonoid contents depending on the different recyclable substrates.

The relationships established between TPC, ABTS, FRAP, DPHH, and flavonoid content in our observations were assessed by a principal components analysis (PCA) (Figure 6). The PCA revealed a particular response of flavonoid content (associated with the F2 axis) compared to the other variables, FRAP, TPC, DPPH, and ABTS (which were associated with the F1 axis).

In order to determine the strength of the relationships between them, a Pearson’s correlation matrix was applied, and it is presented in Table 2. All the significant results showed positive correlations, which highlights the stimulating influences of the treatments. The total phenolic content, in order of p significance (Table 2 in gray), was correlated with DPPH, FRAP, and ABTS. DPPH presented the strongest influence on ABTS (r = 0.86; *p* = 0.0076). Total phenolic content contributes to DPPH and ABTS activity. In addition, the flavonoids content contributes to FRAP, and the total phenolic content contributes to DPPH and ABTS activities.

### 3.6. Protein Content

The dried mushroom powders were analyzed using the macro-Kjeldhal method. The total protein content was measured and recorded in Figure 7, achieving a higher content in T1, the pine sawdust substrate, with significant difference between the three treatments’ ANOVAs (*p* ≤ 0.05). Coconut coir (T2) showed a lower value than T1 but closer to paper waste substrate (T3), which was the lowest one (Figure 7).

### 3.7. Sensory Evaluation and Organoleptic Properties

After data collection, statistical analyses were performed to identify the preferences and acceptability of oyster mushrooms grown on different substrates. A one-way ANOVA and a Tukey test were performed to determine significant differences between treatments, checking the hypothesis test with a *p* < 0.05 for each parameter. Table 3 reports the mean value of each sensory attribute and acceptance for each treatment.

Regarding the sensory parameters, whose comparison is observed in Figure 8, it was found that the three treatments evaluated showed similar attributes, the coconut coir substrate being the one to produce fruiting bodies with better sensory attributes but with no significant differences between treatments. Some parameters are not relevant for the analyses, such as sourness, astringency, or bitterness, since they were assessed generally by “not perceived at all”.

The acceptability test revealed similar results for the three treatments, with little significant differences (Figure 9). For “Taste”, T2 (11.867) obtained a significantly higher score than the other two treatments, T3 was the lowest one. Regarding the “Color”, there are no significant differences. For the “Texture” attribute, T2 (11.967) shows a higher value than the other treatments. The lowest value of “Appearance” (11.133) was expressed for T1, grown on pine sawdust. The overall acceptability positioned the mushrooms grown in coconut coir (T2) as the product of preference. On the contrary, the mushrooms grown on pine sawdust showed the lowest score, although there are no significant differences between the three treatments.

## 4. Discussion

### 4.1. Biological Efficiency, Mushroom Yield and Production Rate

*P. ostreatus*, also known as “oyster mushroom”, “hiratake”, “shimeji”, or “houbitake”, is able to grow in available waste materials. The different waste by-products of lignocellulose composition tested as substrate for the *P. ostreatus* cultivation represented by pine sawdust, coconut coir (fiber mixed with shell), and waste paper were found to be a good support for the growth of the fungus, with the mycelium fully colonizing the substrates at 28 days. A similar mycelium growth rate, however, did not correspond with yield, indicating that the mycelium growth and yield of mushrooms have different requirements [46]. The results showed that mushroom yield is reliant on biological efficiency, as overall biological efficiency determines the mushroom yield, which is in accordance with the findings of [47], displaying that mushroom yield is dependent on biological efficiency. In our study, the highest yield was harvested from paper waste substrate (T3) with the most elevated percentage of biological efficiency, followed by coconut coir (T2), while the lowest was observed in pine sawdust (T1). Similarly, the biological efficiency (BE) also varied significantly among the different substrates used. The performance of oyster growth and yield in the sawdust substrate was minimal, a result similar with the data obtained by [31]. This could be attributed to the fact that the lignocellulosic materials in pine sawdust are generally low in protein content and insufficient for the mushroom’s cultivation [46]. Therefore, the sawdust substrate for mushroom production should undergo a period of composting to breakdown the cellulose and lignin components of the wood in order to release the essential materials for the establishment of mushroom mycelium.

In addition, the mean comparisons (separated using Tukey test) revealed that the biological yield from paper waste was significantly different from the rest of the substrates at a 5% confidence level (Figure 1B). The results of this study are in line with other studies elsewhere (e.g., [6,30]), where paper was identified as an important substrate for significant improvement in the yield of oyster mushroom. The possible justification may be that the paper waste is a high C-content waste material, i.e., PW (C/N = 379) [48], and is accepted as a superior substrate over pine sawdust.

Generally, the present study confirmed that oyster mushrooms can grow on pine sawdust, paper waste, and coconut coir, with varying growth performances. Paper waste was identified as a suitable substrate for oyster mushroom cultivation, since it produced a significantly higher yield, biological efficiency, and production rate compared to the other substrates. The BE, yield, and production rate of both coconut coir and pine sawdust proved similar values, where coconut coir have slightly higher parameters than pine sawdust. Paper waste proved to be better in terms of mycelium density, pin-head formation, and the development of fruiting bodies, and it is a good recommendation as a preferred substrate for oyster mushroom cultivation, serving as a viable solution for the environmental contamination by using the huge paper wastes available. In addition, coconut coir, a contaminant waste product in coconut producing countries, can be used as an alternative substrate given that the growth performance and yield of oyster mushrooms was better than the pine sawdust.

### 4.2. Antioxidant Activity

Oxidation is essential for the living organisms to produce energy for the biological processes. Free radicals are produced in normal and pathological cell metabolism. The uncontrolled production of oxygen-derived free radicals is involved in the onset of many diseases. The antioxidant components are responsible for defending our body against free radicals, and it is known that low levels of antioxidants cause oxidative stress and may damage or kill cells [49]. Numerous fungi were reported to have antioxidant components, higher than in most vegetables and fruits, and concentrated in fruit bodies and both mycelium and culture. For this reason, our study was also conducted to evaluate the antioxidant activity of the oyster mushroom, depending on the waste by-product used as substrate, using spectrophotometrically methodologies with different mechanisms, one that report the scavenging ability on DPPH and ABTS radicals (DPPH and ABTS assays) and the other monitoring the reducing power of compounds as a significant indication of its potential activity [50]. The presence of reducers (i.e., antioxidants) causes the reduction of the Fe^3+^/Ferricyanide complex to ferrous form in the case of the FRAP assay.

The antioxidant activities measured in fungi ethanolic extracts were obtained using the three DPPH, ABTS, and FRAP assays that gave a comparable ranking of antioxidant activity among the substrates used with the highest antioxidant capacity revealed by the coconut coir substrate (T2). So, this waste by-product used as substrate for oyster cultivation showed the highest values of scavenging ability on DPPH and ABTS free radicals with the range of 44% to 65%. The results of the FRAP assay indicate that the significantly highest reducing power inhibition could be identified in the extract of oyster fungi grown in the same substrate represented by coconut coir. This substrate has been found to significantly reduce the power in ferric ions of oyster fungi due to its effect on the total phenolic and flavonoid contents of the fruiting body extracts, which play an important role in antioxidant activities. According to [51], the reducing power might be due to their hydrogen-donating ability, and certain mushrooms contain higher amounts of reduction, which could react with free radicals to stabilize and terminate radical chain reactions.

### 4.3. Total Phenolic Content (TPC)

Phenolic compounds possess a common chemical structure comprising an aromatic ring with one or more hydroxyl substituents that can be divided into several classes, and the main groups of phenolic compounds include flavonoids, phenolic acids, tannins, stilbenes, and lignans [52]. It has been reported that phenolic compounds exhibit antioxidant activity in biological systems, acting as free radical inhibitors, peroxide decomposers, metal inactivators, or oxygen scavengers [53,54]. Phenolic compounds are present in all the mushrooms. These compounds can be pyrogallol, myricetin, caffeic acid, quercetin, and catechin, among others. The fruiting bodies of *Pleurotus* respond dramatically to the chemical composition of the substrate where they grow and develop. The bioactive compounds, including phenolics, can be effectively absorbed by the fruiting bodies of *Pleurotus* [55,56]. Our study could register a significant difference in TPC of *P. ostreatus* grown on various waste by-products substrates. The highest values of TPC were shown in fruiting bodies of the edible fungi obtained from coconut coir containing substrate that exhibited a TPC in the range of 586.60 ± 31.97 mg GAE/g dry weight compared to the lower values of 397.21 ± 14.87 mg GAE/g dry weight on waste paper substrate and followed by 302.95 ± 19.40 mg GAE/g dry weight on the pine sawdust substrate. The changes in the TPC of the oyster fungi fruiting bodies grown in different substrate formulas are explained by the difference in the lignin composition of the substrates. The presence of lignin in substrates reduces the number of some biological active molecules, which is directly linked to a decreased biological activity [57]. This activity also depends on how easily the substrate is decomposed by the mycelium and the quality of nutrients assimilated by the mushrooms [58]. It was observed that the coconut coir substrate significantly increased the TPC and antioxidant activity against DPPH and ABTS+ radicals. It was noticed that the phenolic contents correlate well with DPPH and ABTS assays, with correlation coefficients of 0.8601 and 0.7429, respectively, confirming that phenolic compounds contribute to radical scavenging activity of the fungi extracts, having a redox potential. These results are in agreement with those observed in *Camellia sinensis*, *Annona muricata*, *Zingiber officinale* [59], and *Syzigium aromaticum* and *Allium sativum* [60].

### 4.4. Total Flavonoid Content (TFC)

Flavonoids represent a group of natural substances with variable phenolic structures, considered an indispensable component in a variety of nutraceutical, pharmaceutical, medicinal, and cosmetic applications. The TFC assay was estimated to extract flavonoids, isoflavonoids, and neoflavonoids, or collectively called bioflavonoids. Several studies have demonstrated that these compounds may act as antioxidant by breaking the radical chains into more stable products in liver microsomal membranes, with the ability to protect low-density lipoprotein or LDL from being demolished by heavy metals and macrophages. They also play an important role in providing instinctive protection against oxidative stress and side effects by its contribution with vitamins.

In the present study, total flavonoid content was calculated using quercetin as a standard and the results are expressed as mg of quercetin equivalents per gram of extract. It showed that the fungi extract with the highest total phenolic content also exhibited the highest total flavonoid content (77.54 ± 16.22 mg QE/g) on the growing substrate represented by coconut coir. The changes in TFC under the influence of substrates can also be explained by the different quantity of lignin, in the same way as it was influenced by the TPC. It could be observed that the flavonoid content in the fungi extracts showed a higher correlation with reducing power FRAP assay than with the radical scavenging activity. This could estimate that the flavonoid compounds present in the extracts act as an antioxidant directly through the mechanism of the reduction of oxidized intermediate in the chain reaction.

### 4.5. Protein Content

Protein is an essential nutrient in human life activities, helping the formation, growth, constitution, or repair of human tissues. The fungal ones provide a healthy new protein with a low environmental impact [61]. *P. ostreatus* is generally considered to be a good source of digestible proteins, providing all the essential amino acids required by an adult [62]. Protein content of *Pleurotus* depends on the composition of the substrates and mushrooms species [63]. The protein analysis of the fruiting bodies from each growing substrates indicate that pine sawdust substrate offered the highest protein content (24.02%) for the oyster fungi followed by 15.64% on the coconut coir substrate and 12.04% on the waste paper. Nutritional value, in particular the protein content of *P. ostreatus* cultivated on pine sawdust substrate, was higher than those exhibited by another popular edible mushrooms Pioppino (*A. aegerita*) with 20.5% and Champignon (*A. bisporus*) mushrooms with 13.4% [41], the last one being slightly similar with *P. ostreatus* cultivated in coconut coir and waste paper substrates. The amount of nitrogen supplied in the growth media may be used to control the protein content of the mushroom mycelium. The carbon/nitrogen influences the protein content in the mushroom mycelium [64]. The obtained data corroborate the results of [65], which reported that Oyster (*Pleurotus*) mushrooms are considered to be one of the most efficient producers of food protein, producing 30% of its dry weight. Carbon is readily available from cellulose, hemicelluloses, and lignin, so pine sawdust represents a good source of carbon [66]. Sharma et al. [67] noted that the C/N ratio significantly influences the values obtained from the protein composition of the fungus *P. ostreatus*. The C/N ratio of the substrate is critical to the initial development of the fungus, given the value of carbon for the formation of new cells; a low C/N ratio in the substrate will influence the fungus negatively during the mycelium growth stage.

### 4.6. Sensory Evaluation and Organoleptic Properties

The sensory attributes of *P. ostreatus* cultivated in different substrates were analyzed through a consumer test, assessed by seven sensory attributes and five acceptability parameters. Mushroom samples were hygienically and neatly prepared to be presented for consumption. Panelists gave similar preference scores for all samples from different substrates, which indicated that all were highly satisfactory as judged by appearance, color, taste, and texture, with an overall liking and acceptance for all three. Nonetheless, panelists showed some significant (*p* < 0.05) preferences at taste for the coconut cultivated oyster mushroom, which could be due to the impact of growth substrate on the oyster mushroom fruiting bodies composition, yield, biological efficiency, and nutritional profile [68]. The flavor experienced from eating mushrooms, or any other food, comes from a combination of taste, texture, temperature, spiciness, and aromatic qualities [41].

Our findings showed that consumer acceptability of the fruiting bodies was largely influenced by their sensory characteristics, as well as by their visual appearance. It is well-documented that sensory characteristics such as taste, appearance, freshness, texture, color, and smell are essential motivating factors to lead consumers towards the consumption of food products [69]. Overall, one of the unconventional tools for sensory evaluation of mushrooms is the smell, which is important for both the identification of species and for the oro-sensory sensations one experiences while eating them, among several issues regarding gustation and olfaction [70].

In our study, recorded sensory attributes were similar for the harvested fruiting bodies from three different substrates. As stated in literature, flavor-related characteristics predict the best the consumer preferences for overall eating quality, and therefore, consumer preferences are affected primarily by sensory characteristics [42]. For the present study, aroma, astringency, bitterness, chewiness, sourness, sweetness and umami, flavors found in mushrooms, were evaluated by the panelists. As Du et al. [71] explain, in mushrooms, carboxylic acids contribute to sour taste, while sweetness can be perceived by the presence of sugars, polyols, and several amino acids, and umami taste is elicited by 5′-nucleotides, monosodium glutamate (MSG), and several other free amino acids and nucleotides. As for texture parameters, chewiness was assessed with a high value and so were liked by the panelists. This feature is related to the mushroom state at which fruiting bodies are harvested, and it is related to good texture and flavor which receive high demand from costumers [72]. Astringency was not perceived at all along with bitterness, sweetness, and sourness; its mouthfeel, as the literature states, is an event induced by tannin interaction and the precipitation of salivary proline-rich proteins (PRPs) in the oral cavity [73].

## 5. Conclusions

Our study is conducted to demonstrate the transformation of wastes (a contaminant problem) into a biological substrate for the edible fungi *P. ostreatus.* In our case, waste paper (the third largest industrial polluter of the environment) was the growth substrate with a higher yield value and biological efficiency for the harvested oyster mushrooms. The substrate suitable for the production of fungi with a higher concentration of bioactive compounds (phenolics and flavonoids) that offer a high antioxidant capacity for a better medicinal quality was represented by coconut coir. This is a waste product that generates great pollution to the ecosystem, favors the proliferation of insects and rodents, and affects the lives of the inhabitants of the sectors where this fruit is grown, since they tend to throw it into the rivers and estuaries. Our study could offer an economical biotechnology for the organic waste recycling of lignocelluloses that combines the production of protein-rich food with desirable organoleptic properties, therapeutic benefits, and the reduction in environmental pollution.

## Figures and Tables

**Figure 1 jof-08-00274-f001:**
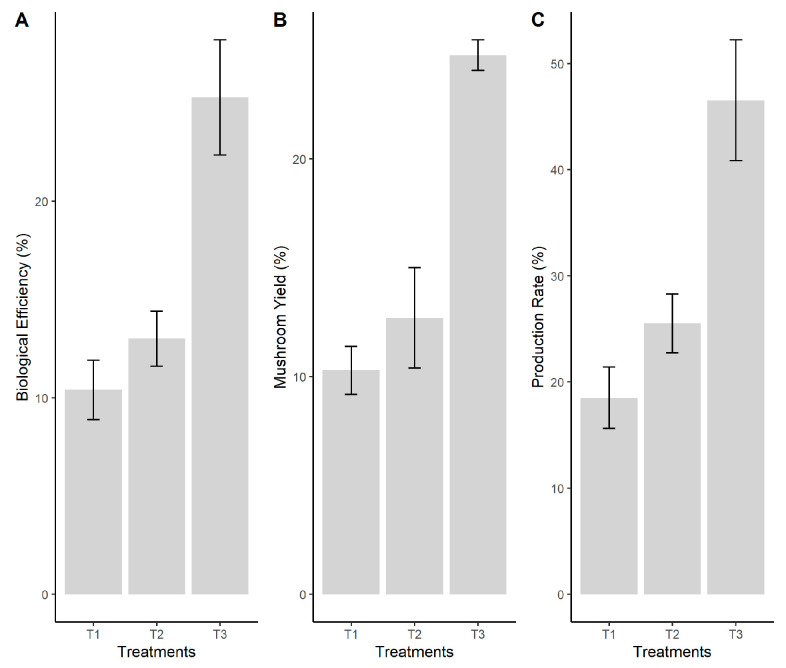
Mushroom Growth analysis in function of the growth media: (**A**) Biological Efficiency (%), (**B**) Mushroom Yield (%), (**C**) Production Rate (%). Legend: T1—pine sawdust; T2—coconut coir; T3—waste paper.

**Figure 2 jof-08-00274-f002:**
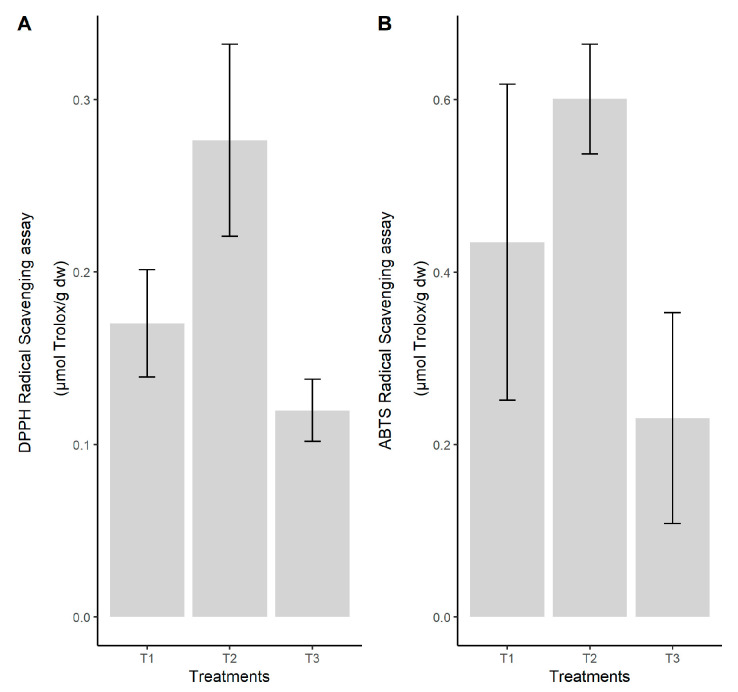
Inhibition percentage (%) and μmol Trolox equivalent per gram of dry weight fungi material, according to the substrate used in the cultivation of *P. ostreatus*: (**A**) DPPH radical scavenging test, (**B**) ABTS radical scavenging test. Legend: T1—pine sawdust; T2—coconut coir; T3—waste paper.

**Figure 3 jof-08-00274-f003:**
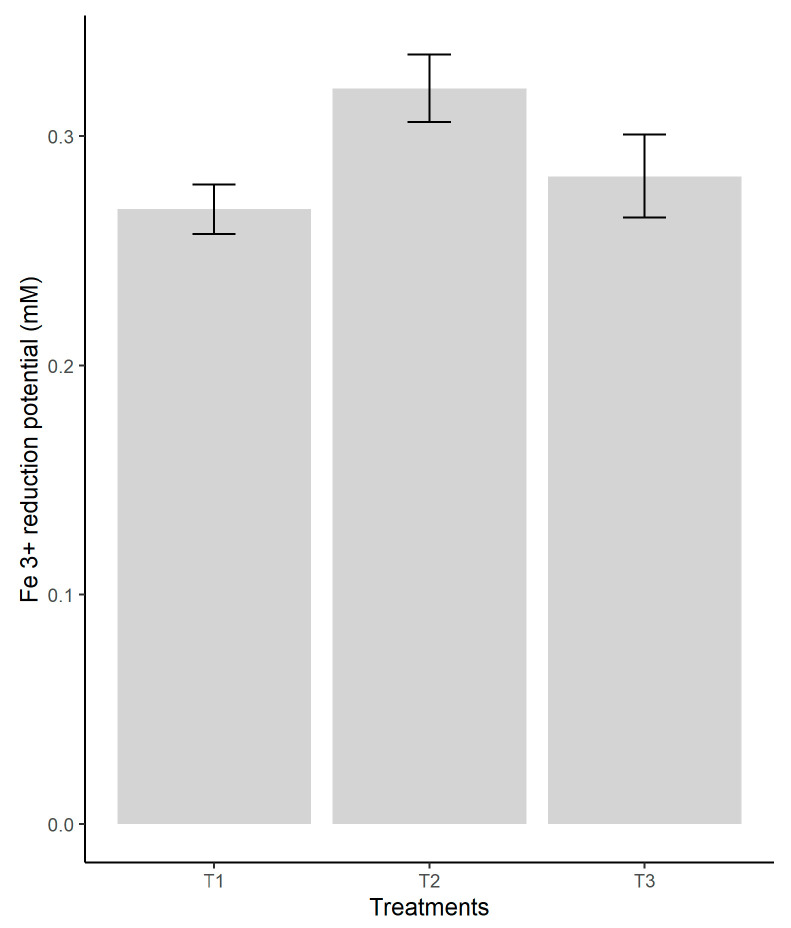
Reducing activity according to the type of substrate used in the cultivation of *P. ostreatus*. Legend: Legend: T1—pine sawdust; T2—coconut coir; T3—waste paper.

**Figure 4 jof-08-00274-f004:**
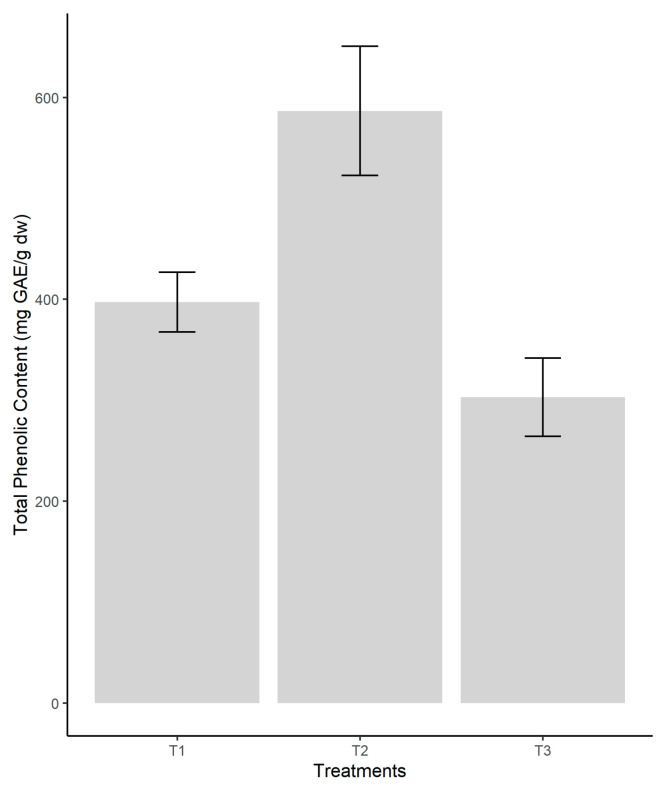
Comparison of the Total Phenolic Content according to the substrate used in the cultivation of *P. ostreatus.* Legend: T1—pine sawdust; T2—coconut coir; T3—waste paper.

**Figure 5 jof-08-00274-f005:**
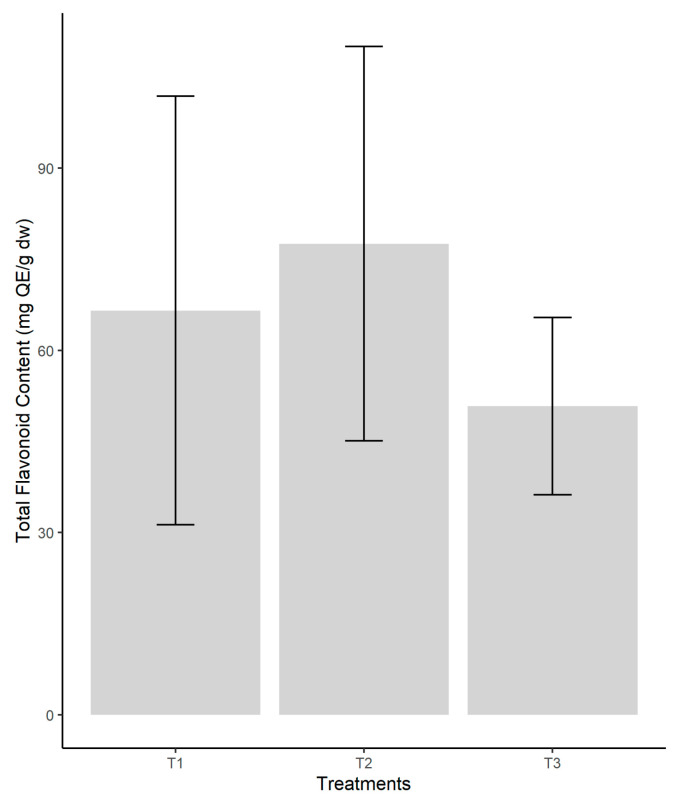
Comparison of the total flavonoid content, according to the type of substrate used in the cultivation of *P. ostreatus*. Legend: T1—pine sawdust; T2—coconut coir; T—waste paper.

**Figure 6 jof-08-00274-f006:**
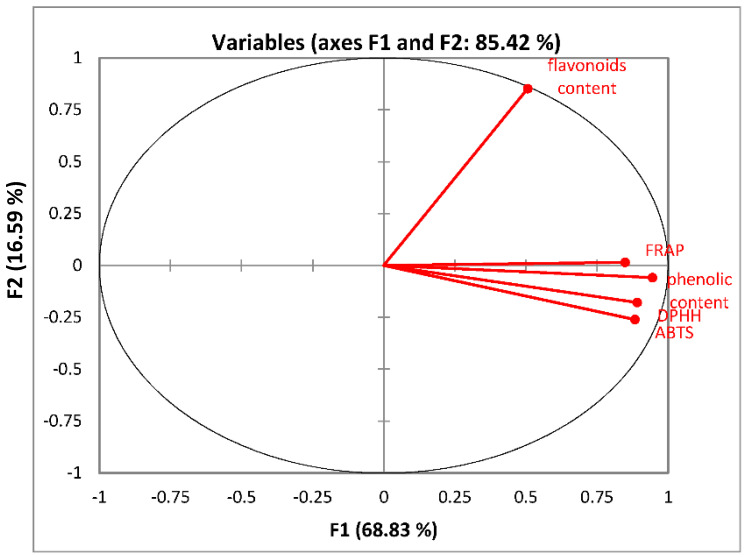
Principal Components Analysis (PCA) between total phenolic content, flavonoid content, and the antioxidant properties DPPH, ABTS, FRAP. Legend: DPPH—radical scavenging test; ABTS—radical scavenging test; FRAP—ferric reducing antioxidant power.

**Figure 7 jof-08-00274-f007:**
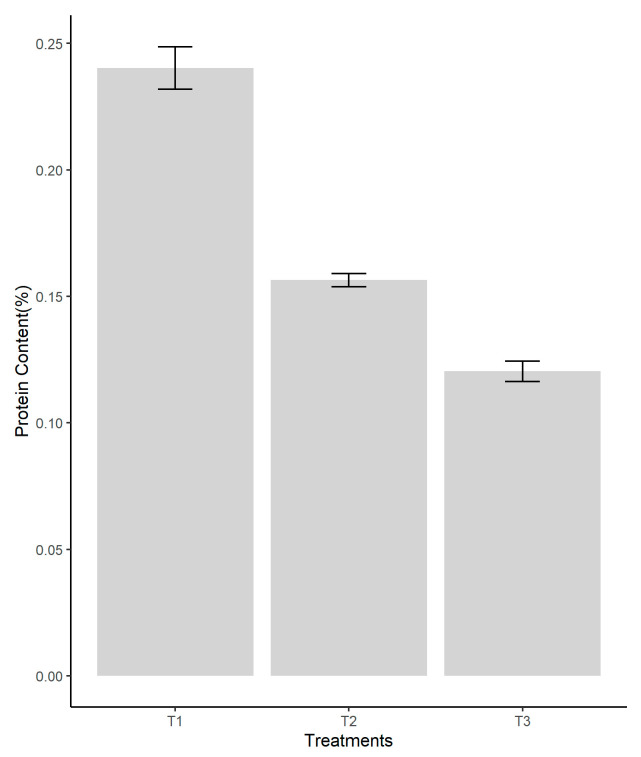
Protein content (%) according to the substrate used in the cultivation of *P. ostreatus*. Legend: T1—pine sawdust; T2—coconut coir; T3—waste paper.

**Figure 8 jof-08-00274-f008:**
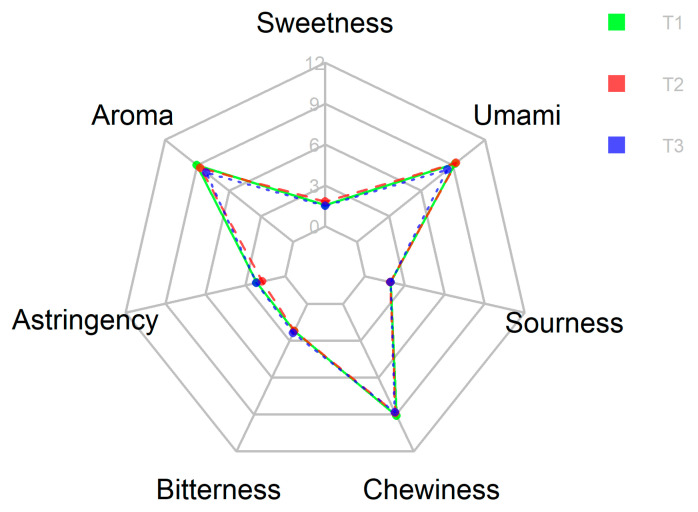
Radar chart for sensory attributes of oyster mushroom grown on different substrates. Legend: T1—pine sawdust; T2—coconut coir; T3—waste paper.

**Figure 9 jof-08-00274-f009:**
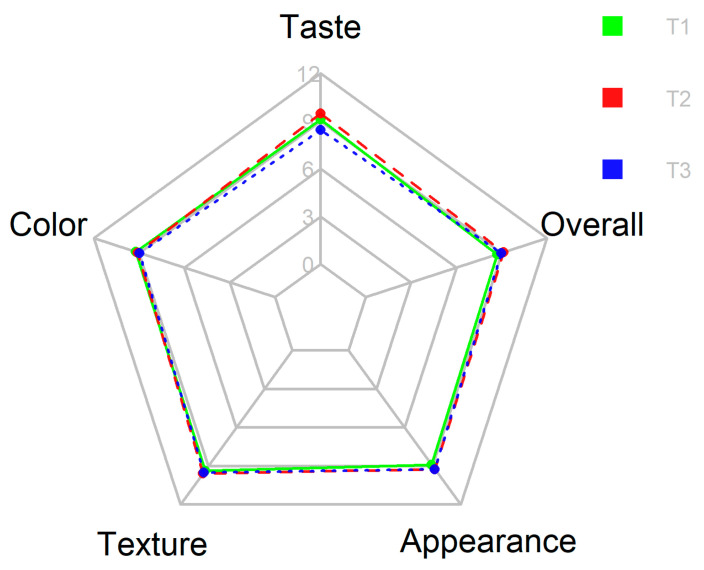
Radar chart for acceptability items for oyster mushroom grown on different substrates. Legend: T1—pine sawdust; T2—coconut coir; T3—waste paper.

**Table 1 jof-08-00274-t001:** Sensory attributes and acceptability parameters evaluated in the consumer test. The fruiting bodies were assessed by seven sensory attributes and five acceptability parameters. The 15-cm labelled scale (LMS) was used to conduct the assessment.

Sensory Attributes	Scale
Sweetness Aroma Astringency Bitterness Sourness Umami Chewiness	0 = not perceived at all 15 = strongly perceived
**Acceptability Parameters**	**Scale**
Appearance Flavor Color Texture Overall Acceptability	0 = bad 15 = excellent

**Table 2 jof-08-00274-t002:** Correlation matrix between the phenolic content (TPC), flavonoid content (FC), radical scavenging test (ABTS), ferric reducing antioxidant power (FRAP), and free radical scavenging (DPPH).

Variables	TPC	FC	ABTS	FRAP	DPHH
TPC		0.2798	**0.0218**	**0.0096**	**0.0029**
FC	0.40		0.4976	0.3074	0.3807
ABTS	**0.74**	0.26		**0.0345**	**0.0076**
FRAP	**0.80**	0.38	**0.70**		0.1161
DPHH	**0.86**	0.33	**0.81**	0.56	

Legend: bold values are the significant results, Pearson correlation coefficients r—lower white half, *p*-values—upper grey half.

**Table 3 jof-08-00274-t003:** Sensory attributes and acceptability parameters evaluated in the consumer test.

Treatment	Sensory Attributes		
	T1	T2	T3
Sweetness	1.933	2.267	1.867
Aroma	11.267	10.867	10.167
Astringency	2.667	2.167	2.700
Bitterness	2.767	2.766	2.967
Chewiness	11.367	11.066	11.000
Sourness	2.400	2.367	2.400
Umami	11.500	11.600	10.600
**Treatment**	**Acceptance**		
	**T1**	**T2**	**T3**
Taste	11.400	11.867	10.567
Color	11.533	11.366	11.233
Texture	11.700	11.967	11.866
Appearance	11.133	11.567	11.566
Overall	10.867	11.400	11.200

Legend: T1—pine sawdust; T2—coconut coir; T3—waste paper.

## Data Availability

Not applicable.

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
