# Peer review of "The Edible Gray Oyster Fungi *Pleurotus* *ostreatus* (Jacq. ex Fr.) P. Kumm a Potent Waste Consumer, a Biofriendly Species with Antioxidant Activity Depending on the Growth Substrate"

_jof, 2022, doi:10.3390/jof8030274_

Round 1
Reviewer 1 Report
Please see the comments in attached file.

Author Response
We would like to thank the reviewer for his or her appreciations on this work. We have addressed the answers to his or her questions below, in red, point by point.
Point 1: Line 20 “Was”
Response 1: We changed it with “It was”
Point 2: Line 86 “Line no. 96 0nward may be place here”
Response 2: We think that the line no. 96 can’t be add to the place where reviewer asked because the phrase it refers to the edible Pleurothus species generally. The line 96 it refers strictly to Pleurotus ostreatus
Point 3: Line 96 “Pleurothus ostreatus”
Response 3: we removed the bold
Point 4: Line 106 “add few more relevant information and reduce the introduction part”
Response 4: we added new information concerning Pleurotus value
Point 5: Line 118 “Whole section need to be revised and precise”
Response 5: we revised the section
Point 6: Line 178 “1.0000 g”
Response 6: we changed
Point 7: Line 184 “keep one heading. this can be deleted”
Response 2: we deleted the line
Point 8: Line 590 “Reduce the conclusion and summarize in brief”
Response 2: we revised and reduced the conclusion section
Point 9: Line 596 “P.” changed
Response 9: we changed according to the reviewer.
Reviewer 2 Report
In the manuscript by Raluca et al., entiteled “The edible gray oyster fungi Pleurotus ostreatus (Jacq. ex Fr.) P. Kumm a potent waste consumer, a biofriendly species with antioxidant activity depending on the growth substrate” was evaluated the possible use of different agricultural wastes (i.e. pine sawdust, coconut coir, and paper waste), representative of the pollution in Ecuador, as suitable opportunities to reduce pollution by recycling them as growing substrates for mushrooms with desirable organoleptic, phytochemical properties and antioxidant activity, with ultimate benefits for human health and the protection of the planet. In this context, although the introduction is clear and the results obtained are interesting, the present version of the manuscript is not acceptable for publication at this step and needs major revisions.
-Page2, lines 96-99. The authors need to add more information on the properties of edible mushrooms such as Pleurotus ostreatus used in this study. Indeed, in recent literature it was reported that edible mushrooms, such as Pleurotus ostreatus, are a source of ribotoxin-like proteins (RL-Ps) a novel family of enzymes capable of inhibiting protein synthesis with numerous biotechnological activities (e.g. cytotoxic, antifungal and antiviral activity). See for example a recent review on this issue (Ragucci et al., Toxins (Basel). 2021;13(4):263).
- Page 6, lines 260-265 and related results: Mushrooms contain relatively large amounts of non-protein nitrogen, considering the chitin of cell and both free amino acids and nucleic acids. The nitrogen attributed to proteins corresponds to 60-77% in mushrooms, hence, the universally used NP factor of 6.25 is too high for mushrooms and should not be used. Considering as above, NP conversion factors are very specific for each species and in case of mushrooms, NP conversion factor is 4.38; as previously reported (Landi et al. Sci Food Agric. 2017;97(15):5388-5397). Please, consider this reference and change accordingly in both materials and methods and results, adding also a comparison with other commercial mushrooms, e. g. Pioppino (Cyclocybe aegerita, synonym Agrocybe aegerita) and Champignon (Agaricus bisporus).
- In the session “Materials and Methods” the authors should add a new paragraph named ‘Statistical analysis’ where they summarize the statistical methods used.
-Revise all Figures, especially the resolution. For graphs, if possible use the same character of the text (e.g. font size) and a proper dimension, higher resolution maintaining image proportion;
- Page 4, line 176: change ‘40˚’ by ’40 °C’;
- Page 4, line 178: change ‘1.0000 g’ by ‘1.0 g’;
- Page 4, line 181: change ‘ml’ by ‘mL’. Check this in overall manuscript;
-Page 4, lines 191-192: change ‘ul’ by ‘µL’. Check this in overall manuscript;
-Page 4, line 192: change ‘Tris-HCl’ by ‘Tris•Cl’;
-References: revise this section according to the journal guidelines: e.g. use bold for indicating the year. Check this in overall references.
Author Response
We would like to thank the reviewer for his or her appreciations on this work. We have addressed the answers to the questions below, in red, point by point.
Point 1: Page2, lines 96-99. The authors need to add more information on the properties of edible mushrooms such as Pleurotus ostreatus used in this study. Indeed, in recent literature it was reported that edible mushrooms, such as Pleurotus ostreatus, are a source of ribotoxin-like proteins (RL-Ps) a novel family of enzymes capable of inhibiting protein synthesis with numerous biotechnological activities (e.g. cytotoxic, antifungal and antiviral activity). See for example a recent review on this issue (Ragucci et al., Toxins (Basel). 2021;13(4):263).
Response 1: We added this reference to our manuscript. Thank you.
Point 2: Page 6, lines 260-265 and related results: Mushrooms contain relatively large amounts of non-protein nitrogen, considering the chitin of cell and both free amino acids and nucleic acids. The nitrogen attributed to proteins corresponds to 60-77% in mushrooms, hence, the universally used NP factor of 6.25 is too high for mushrooms and should not be used. Considering as above, NP conversion factors are very specific for each species and in case of mushrooms, NP conversion factor is 4.38; as previously reported (Landi et al. Sci Food Agric. 2017;97(15):5388-5397). Please, consider this reference and change accordingly in both materials and methods and results, adding also a comparison with other commercial mushrooms, e. g. Pioppino (Cyclocybe aegerita, synonym Agrocybe aegerita) and Champignon (Agaricus bisporus).
Response 2: We have made the required changes.
Point 3: In the session “Materials and Methods” the authors should add a new paragraph named ‘Statistical analysis’ where they summarize the statistical methods used.
Response 3: we added the new paragraph.
Point 4: Revise all Figures, especially the resolution. For graphs, if possible use the same character of the text (e.g. font size) and a proper dimension, higher resolution maintaining image proportion;
Response 3: we revised the figures.
Point 4: Page 4, line 176: change ‘40˚’ by ’40 °C’;
Response 4: We have made the required changes.
Point 5: Page 4, line 178: change ‘1.0000 g’ by ‘1.0 g’;
Response 5: We have changed.
Point 6: Page 4, line 181: change ‘ml’ by ‘mL’. Check this in overall manuscript;
Response 6: We have made the required changes.
Point 7: Page 4, lines 191-192: change ‘ul’ by ‘µL’. Check this in overall manuscript;
Response 7: We have made the required changes.
Point 8: Page 4, line 192: change ‘Tris-HCl’ by ‘Tris•Cl’;
Response 8: We have made the required changes.
Point 9: References: revise this section according to the journal guidelines: e.g. use bold for indicating the year. Check this in overall references.
Response 9: we revised the references according the journal guidelines.
Round 2
Reviewer 2 Report
Thanks for taking my suggestions